# SYMPTOMIFY: Transforming Symptom Annotations with Language Model Knowledge Harvesting

**Bosung Kim** and **Ndapa Nakashole**
Computer Science and Engineering
University of California, San Diego
La Jolla, CA 92093
bosungkim@ucsd.edu, nnakashole@eng.ucsd.edu

## Abstract

Given the high-stakes nature of healthcare decision-making, we aim to improve the efficiency of human annotators rather than replacing them with fully automated solutions. We introduce a new comprehensive resource, SYMPTOMIFY, a dataset of annotated vaccine adverse reaction reports detailing individual vaccine reactions. The dataset, consisting of over $800k$ reports, surpasses previous datasets in size. Notably, it features reasoning-based explanations alongside background knowledge obtained via language model knowledge harvesting. We assess data quality, and evaluate performance across various methods and learning paradigms, paving the way for future comparisons and benchmarking.[1]

## 1 Introduction

Drug safety monitoring systems like Vigibase (Lindquist, 2008) and VAERS[2] are essential for evaluating the safety of medications and vaccines. These systems enable *anyone* to report drug reactions, facilitating early detection of vaccine safety issues. For example, if a mother observes vomiting and fever in her two-year-old after the chickenpox vaccine, she may be curious about the frequency of similar reactions reported by others. VAERS' online browser can provide the answer. Being able to provide such answers involves normalizing the raw patient reports in order to address variations caused by polysemy, abbreviations, and other factors present in these informal and potentially noisy reports.

VAERS employs trained staff to review and "code" each submitted report, assigning formal medical terms to the symptoms experienced by the patient. VAERS terms come from the standardized medical terminology known as the Medical

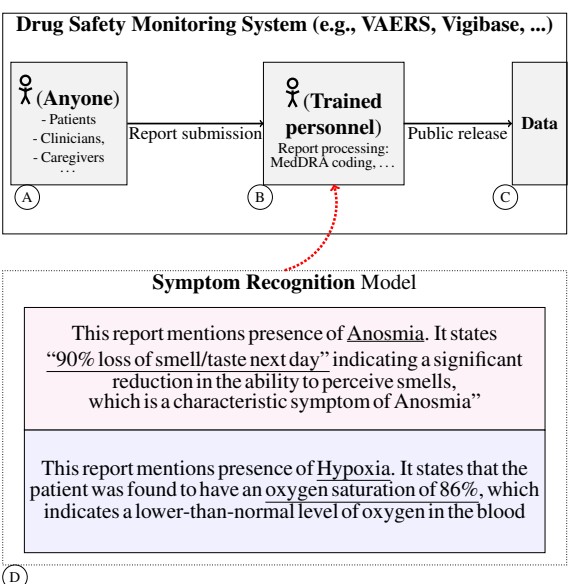

Figure 1: **A to C** illustrate the stages of drug safety monitoring systems like VAERS. We aim to improve the efficiency of skilled human coders in step **B** by facilitating the development of systems that identify symptoms and offer explanations to aid coders, as demonstrated by the model-generated output in step **D**.

Dictionary for Regulatory Activities (MedDRA) (Brown et al., 1999; Shimabukuro et al., 2015). To illustrate the coding process, consider the report:
***R1**: "I had a 90% loss of smell/taste the next day. Also felt irritable and nervous but manageable."*
This report is annotated with these MedDRA terms: *Ageusia, Anosmia, Irritability, Nervousness.*
Our aim is to support the decision-making of human coders during annotation, by supporting the development of symptom recognition models that can predict and explain identified symptoms (Figure 1). Given the high-stakes nature of this field, where data interpretation can carry life-or-death consequences, fully automated solutions are not yet sufficiently accurate. Specifically, we present a resource of annotated reports detailing people's reactions to vaccines, and enriched in several ways. In summary, our contributions are as follows:

---

[1]The dataset and code are available at https://github.com/bosung/SYMPTOMIFY.

[2]Vaccine Adverse Event Reporting System https://vaers.hhs.gov/data.html

1. **SYMPTOMIFY Dataset:** We present a symptom recognition dataset derived from VAERS informal medical reports with over 800k entries. Beyond its scale, we enhance the dataset by adding explanations and background knowledge substantially boosting its utility for advancing systems that aid human coders in annotation.

2. **Annotation Explanations:** We enrich SYMPTOMIFY with annotation explanations, carefully harvested from LLMs. We showcase their usefulness on the newly released Falcon language model.

3. **Background Knowledge Augmentation:** We augment SYMPTOMIFY with background knowledge about symptoms, improving performance, particularly for rare symptoms.

4. **Performance Evaluation:** We implement and evaluate various baselines across different methods and learning paradigms, facilitating future comparisons and benchmarking.

## 2 Related Work and Existing Datasets

Our work is related to the old task of Named Entity Recognition (NER), which is the task of mapping mentions of entities in text to standardized entity names (Sutton and McCallum, 2004; Ratinov and Roth, 2009; Ritter et al., 2011; Hoffart et al., 2011; Cao et al., 2021). Our work goes beyond recognizing entities as we wish to support *human decision making*. Additionally, our work is complementary to work on Explainable Artificial Intelligence (XAI) (Doshi-Velez and Kim, 2017; Ribeiro et al., 2016), and can be used to build more explainable models. Furthermore, there are efforts to enhance open-source LLMs by extracting instruction-output pairs from massive, paid-API LLMs. The goal is to gather valuable data that can be utilized for instruction tuning of open-source models. This method is particularly beneficial for projects that lack the financial resources to hire people for human feedback, thereby democratizing access to advanced LLM capabilities (Peng et al., 2023).

Closer to our work are prior efforts on creating datasets for medical entity recognition from diverse sources, including death certificates (Goeuriot et al., 2017), scientific publications (Verspoor et al., 2012; Mohan and Li, 2019), and electronic health records (EHRs) (Suominen et al., 2013). While these corpora contribute to medical entity recognition, they

| Dataset | Source | # Symptoms | Size |
|---|---|---|---|
| **SYMPTOMIFY** (ours) | VAERS | **11,472** | **871,373** |
| COMETA (Basaldella et al., 2020) | Reddit | 7,645 | 20,000 |
| MedRed (Scepanovic et al., 2020) | Reddit | 18 | 2,000 |
| RedMed (Lavertu and Altman, 2019) | Reddit | 2,978 | n/a |
| Twitter TwiMed (Alvaro et al., 2017) | Twitter | 3,144 | 2,000 |
| CADEC (Karimi et al., 2015) | AskaPatient | 6,754 | 1,253 |
| Twitter ADR (Nikfarjam et al., 2015) | Twitter | 1,280 | 1,784 |

Table 1: SYMPTOMIFY surpasses the scale of previous related datasets for entity recognition in informal health reports. Additionally, SYMPTOMIFY contains annotation explanations and background knowledge about symptoms.

differ from the focus of our work, which centers around user-generated text.

Table 1 summarizes datasets closely related to ours, with many, like ours, targeting pharmacovigilance, and unlike ours, originating from social media platforms such as Twitter and Reddit. Like our work, these efforts aimed to identify symptoms mentioned in unstructured and informal textual sources. SYMPTOMIFY surpasses previous datasets in size, and notably, it incorporates annotation explanations and provides background knowledge about symptoms.

## 3 LM Knowledge Harvesting

Access to the most powerful LMs is often limited to paid APIs with query restrictions. In dynamic settings, such as continuous MedDRA report annotation, rapid query response is crucial. Therefore, we propose to extract a portion of LM knowledge to enrich an *existing dataset*. This approach allows greater control, transparency, and balance between performance, cost-efficiency, and autonomy.

Enriching existing datasets meets a widespread need as under-specified datasets are common. A prominent example comes from Question Answering tasks where only few datasets are fully labeled with question, answer, passage, and span.

In our work, key steps to upgrade a dataset via LM knowledge harvesting include: (i) *Identifying Shortcomings:* Assess the dataset to determine missing elements and areas needing improvement. (ii) *Creating Good Prompts:* Design prompts using existing dataset examples to minimize potential ambiguities. (iii) *Checking Data Quality:* Implement strategies to assess LLM output quality, ensuring it is accurate, useful, and safe. (iv) *Maintaining Detailed Records:* Document the harvesting process, including prompts, LLM configurations, and dates, for improved transparency and reproducibility.

## 4 The SYMPTOMIFY Dataset

With VAERS[3] as the starting dataset, we used LM knowledge harvesting to create SYMPTOMIFY. The dataset spans three years (2019-2021) and includes $839,215$ reports, incorporating symptom texts, relevant MedDRA terms, and associated metadata like age, sex, and vaccine type.

### 4.1 Quantitative Analysis of MedDRA coders

VAERS uses certified MedDRA coders to assign formal medical terms to symptoms. We evaluated the quality of these annotations via crowdsourcing on Amazon Mechanical Turk (AMT). We first randomly selected $1,000$ symptoms and then selected a corresponding patient report (symptom text) mentioning that specific symptom. A symptom was considered correctly annotated if two out of three evaluators agreed.[4] Based on this accuracy metric, VAERS staff correctly labeled $93.4\%$ of the $1,000$ tested symptoms, highlighting the high quality of their annotations. Still, this figure might underestimate the true accuracy, as most challenging examples were medical test results not stated in patient reports (such as *C-reactive protein levels* or *metabolic function tests*). The agreement rate among evaluators was moderately strong ($0.55$ Fleiss' Kappa[5]), similar to related annotation tasks reported in the literature which showed agreements in the range: $0.55 - 0.62$ (Wadhwa et al., 2022; Nye et al., 2018; Deleger et al., 2012).

**ChatGPT as the $4$-th annotator.** ChatGPT (based on GPT-3.5) served as a fourth annotator alongside three human coders, working on the same $1,000$ tasks. The provided prompt was: *"Does the following patient report mention '{SYMPTOM}': '{REPORT}'?".* ChatGPT agreed with MedDRA in $86.7\%$ of the cases. However, this figure might overestimate ChatGPT's symptom recognition capacity, as this binary task is significantly simpler than unguided symptom identification. We further probed ChatGPT's abilities through tests more authentically reflecting symptom recognition.

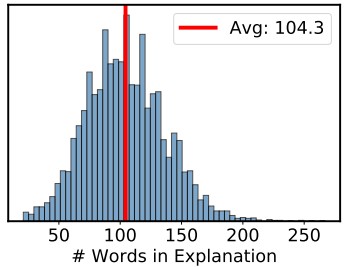

Figure 2: Distribution of explanation lengths in number of words.

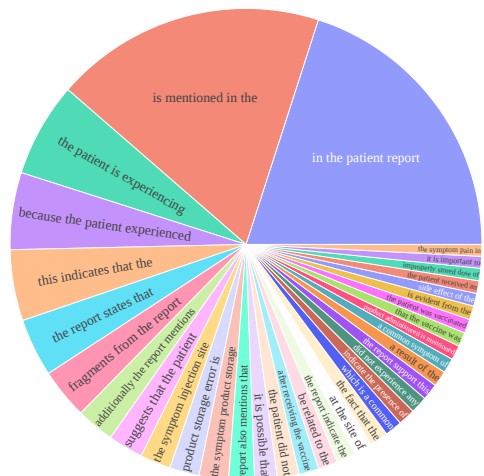

Figure 3: Visualizing the top 30 most frequent 4-grams in explanations. These phrases often indicate that the explanations quote spans from the patient reports.

### 4.2 Harvesting Annotation Explanations

We augment VAERS data by incorporating explanations to compensate for the absence of span-level annotations. Using GPT-3.5, we generated explanations to add to the dataset. The prompt used was: *"Explain why the patient report mentions '{SYMPTOM}', quoting report fragments where possible: '{REPORT}'."* ChatGPT's responses were then used as explanations. These explanations offer extended context beyond the report's explicit content, like identifying *Hypoxia* from a report referencing an "oxygen saturation of 86%", correctly interpreting it as low blood oxygen. By directly quoting from the patient report, the explanations also provide span-level information.

**Qualitative Analysis of Explanations.** Figures 2 and 3 illustrate the distribution of explanation lengths and the 30 most recurrent 4-grams within the explanations, respectively. Often, these frequently appearing phrases refer to the symptom mentions in patient reports, including phrases like "is mentioned in the," "the report states that," "frag-

---

[3]VAERS is jointly managed by the U.S. Centers for Disease Control and Prevention (CDC) the U.S. Food and Drug Administration (FDA)

[4]Details of the annotation task are provided in Appendix A.1.

[5]Overall, the annotation process cost $723 (US dollars) and involved 201 workers.

| Vaccine | Symptom | Explanation Fragment |
|---|---|---|
| FLU4 | Dizziness postural | This **is evident from the** following fragment of the report: "I became extremely dizzy when I tried to stand. |
| COVID19 | Lymphadenopathy | This **is evident from the** statement "My lymph nodes were swollen." |
| FLUA3 | Throat tightness | The following **fragments from the report** indicate the presence of the symptom: - "throat feeling like its about to close" - "swelling and lump in throat on right side" |
| HEPA | Thirst | The following **fragments from the report** support this explanation: "He then began drinking large amounts of water throughout the day and even waking at night to ask for a drink multiple times." |
| FLUA4 | Arthralgia | The report states "Shoulder pain" which is **a common symptom of** arthralgia. |
| FLUR4 | Dyspnoea | The symptom "Dyspnoea" is mentioned in the patient report because the patient is experiencing difficulty breathing, which is **a common symptom of** dyspnoea. |

Table 2: Example explanation fragments featuring some of the most common 4-grams, demonstrating direct quotation of spans from the reports and the use of background knowledge about symptoms when necessary.

ments from the report," etc. Table 2 shows fragments from example explanations that feature these common 4-grams, illustrating the direct quotation of spans from the reports and the use of background symptom knowledge.

**Quantitative Analysis of Explanations: Honesty and Helpfulness.** We evaluated the explanations on two primary dimensions: 1) Honesty: *Is the explanation factually accurate, without fabrications or distortions?* and 2) Helpfulness: *Is the explanation correct and beneficial in clarifying why a symptom is mentioned?*. We randomly picked $1,000$ explanations and employed a three-choice task for assessment: A) The explanation is both honest and helpful; B) The explanation is honest but not detailed enough to be helpful; and C) The explanation is neither honest nor helpful (i.e., incorrect explanation). Each task was evaluated by three annotators, with a label deemed accurate if at least two annotators agreed on it. Our findings show that $91.3\%$ of the explanations were honest and helpful, $7.5\%$ were honest but not helpful, and $1.2\%$ were neither honest nor helpful. For this relatively complex task, we hired three graduate students for annotation. The agreement rate was moderately strong, with a Fleiss's Kappa of $0.60$.

### 4.3 Harvesting Background Knowledge

Sometimes, recognizing symptoms in a report requires understanding beyond what's written. For example, a report might state "BP is 173/109" without mentioning hypertension directly. But, given that hypertension is defined as blood pressure above 140/90, the report implies this condition. This kind of understanding demands external knowledge. Thus, we aimed to augment SYMP-TOMIFY with background knowledge about symptoms.

As a basic form of background knowledge we link each unique symptom in SYMPTOMIFY with

| Symptom | LLM Definition | UMLS Definition |
|---|---|---|
| Melaena | Blood in the stool, typically caused by gastrointestinal bleeding. | The black, tarry, foul-smelling feces that contain degraded blood. |
| Ischaemia | Lack of blood flow to a tissue or organ. This may be due to obstruction or a problem with the blood vessels. | A decrease in blood supply caused by blockage of blood vessel. |

Table 3: Examples of symptom definitions derived from both UMLS and a Language Model (LM). The comparison with UMLS definitions underscores the LM's capability to generate high-quality, medical term definitions.

its definition. We harvested symptom definitions using the LLM[6]. Here one could also use knowledge graphs such as the Unified Medical Language System (UMLS) (Bodenreider, 2004), but we found coverage to be incomplete. In our experiments, these definitions were used in zero-shot learning baselines. But, there are many other potential applications that future research could explore. Currently, SYMPTOMIFY's symptom background knowledge primarily consists of definitions. However, there are avenues to expand this knowledge further. For example, more complex relational structured data, such as known relationships between symptoms, could be extracted from large LMs in future endeavors.
.

**Qualitative Analysis of LLM Definitions.** In Table 3, we display two example definitions generated by the LM, comparing them with UMLS definitions to highlight the LM's proficiency in producing high-quality explanations. Certain terms used in drug safety monitoring systems may have dual meanings, creating ambiguity both within and outside the medical context. To bolster a medically-focused comprehension, particularly regarding adverse effects, we recommend integrating the phrase "adverse effect" into the definition prompt, and sup-

---

[6]We used, from the GPT series, the `text-davinci-001` version.

plementing the prompt with example reports from VAERS that mention the symptom in a few-shot style. This method mitigates potential misunderstandings from multiple interpretations of a term.

**Quantitative Analysis of LLM Definitions.** To quantitatively evaluate the definitions, we randomly selected $1,000$ definitions and assigned three crowd-sourced annotators to rate each definition's accuracy. A definition was deemed accurate and useful if it gained agreement from at least two annotators. The evaluation results indicated that $96.4\%$ of the definitions were accurate. The inter-rater agreement, Fleiss' Kappa, was 0.34, which indicates a fair degree of consensus among the raters.[7]

## 5 Experimental Study

**Data.** Our experiments use the following data splits: Training set with $671,373$ instances, validation set with $83,921$ instances, and test set with $83,921$ instances. We further segment the test set into three subsets for more granular analysis: i) **Full**: The entire test set. ii) **UMLS-mapped**: Exclusively contains symptoms that can be mapped to UMLS, identified via exact string matching, for fair comparison with previous UMLS-focused studies, specifically Metamap. iii) **Rare**: Only includes 'long-tail' symptoms, those occurring in 50 or less reports, representing over 80% of all symptoms. More details of the experimental configuration can be found in the appendix.

**Task.** To set the initial benchmarks on our dataset, we started by focusing on the accuracy of symptom recognition, without considering the added information from explanations or symptom background knowledge. We represented each data point as a pair $(x, \mathcal{S}_i)$, where $x$ is a patient report and $\mathcal{S}_i = \{s_1, ..., s_n | s_i \in \mathcal{S}\}$ is the set of symptoms mentioned in $x$. The goal is to predict the MedDRA names of symptoms mentioned in the input report $x$. We explore a variety of baseline methods across different learning paradigms:

1. **String Matching:** Involves matching symptom names directly with the report text.

2. **Symptom Embedding:** Represents symptom names and report texts as dense vectors for matching.

---

[7]The annotation process engaged $1,437$ workers and amounted to a cost of $522.

| Model | Full test | | UMLS-mapped | |
|---|---|---|---|---|
| | Macro F1 | Micro F1 | Macro F1 | Micro F1 |
| Exact String Matching | 8.87 | 30.38 | 10.41 | 32.98 |
| MetaMap | - | - | 3.74 | 10.65 |
| Symptom Embedding | 0.72 | 3.32 | 0.87 | 5.87 |

Table 4: Performance of string matching and symptom embedding methods. These methods do not utilize the training data.

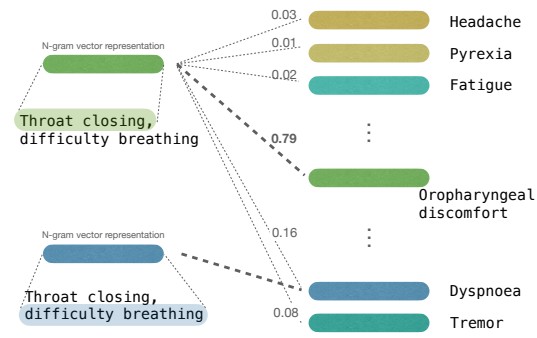

Figure 4: An illustration of symptom embedding baseline. It is similar to span-based string matching, but performs matches using continuous representations rather than discrete symbols.

3. **Pretrain-Finetune:** Uses pre-trained language models fine-tuned on our dataset.

4. **Prompting:** Leverages prompts to generate responses from language models in zero-shot and few-shot settings.

### 5.1 String Matching Baselines

Our initial baselines attempt to align symptom mentions with MedDRA terms without considering context or utilizing training data. We employ two methods: (i) **MetaMap**: A National Library of Medicine-developed tool that maps text spans to symptom names in the UMLS dictionary. For fairness, we only evaluated MetaMap on the test set subset that could be UMLS-mapped. (ii) **Exact String Matching**: A stringent method that requires precise string matches, offering a stricter comparison than MetaMap's span matching.

The results in Table 4 demonstrate the task's complexity, as these string matching methods perform poorly. The findings underline the shortcomings of simplistic approaches in effectively handling this task.

### 5.2 Symptom Embedding Baseline

As shown in Figure 4, our symptom embedding method uses dense retrieval from a frozen language model. Symptom vector representations

are created from symptom definitions in SYMP-TOMIFY. For each symptom $s_i$, we construct an embedding $\mathbf{h}_{s_i}$ using its name and definition. Similarly, we generate n-gram span embeddings for each n-gram in the patient report using ClinicalBERT (Alsentzer et al., 2019) and the span embedding method by Ujiie et al. (2021). We then compare each symptom embedding with all n-gram embeddings using the scoring function $f(s_i, x_{i,i+(n-1)}) = \mathbf{h}_{s_i} \cdot \mathbf{h}[x_i, x_{i+(n-1)}]$. We recall spans with scores exceeding a certain threshold, determined by the best validation score. This approach is similar to zero-shot entity linking by Wu et al. (2019). In our experiments, with $n = 2$, the embedding method did not surpass the string matching method, suggesting that further training of embeddings may be needed (see Table 4).

### 5.3 Pretrain-Finetune Baselines

Given the substantial data available in SYMP-TOMIFY, we investigated fine-tuning baselines.

**Classification Approach.** In a multi-label classification context, we experimented with various pre-trained language models including BERT (Devlin et al., 2019), BART (Lewis et al., 2020), and ClinicalBERT (Alsentzer et al., 2019). We used BERT's classification paradigm, using the `[CLS]` vector for the sentence representation and a classifier layer. Additionally, we implemented a Convolutional Neural Network (CNN) on top of BERT's last hidden states (BERT + CNN) to capture more granular features from the contextualized representations, following the approach by Safaya et al. (2020).

**Generative Approach.** In the context of symptom generation, we adopt the generative entity retrieval approach of GENRE, utilizing a transformer-based encoder and decoder (Cao et al., 2021; Lewis et al., 2020). In this framework, the model generates a sequence of symptom names as the target output, rather than individual entity classification. To retrieve multiple symptoms, GENRE needs annotated spans referring to each symptom. However, as mentioned, VAERS annotations lack symptom mention spans; we only know if a symptom was mentioned or not. Thus, we generate a target sequence as a comma-separated list, e.g., given the input report: *"I have muscle pain and fever"*, the target sequence is: *"Myalgia, Pyrexia"*. We consider baselines using BART and T5 as the pretrained models.

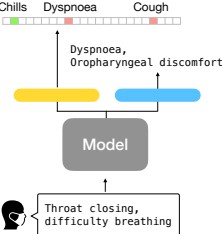

Figure 5: The joint learning method combines discriminative and generative approaches in the pretrain-finetune paradigm. The classifier head (yellow) and the generation head (blue) are illustrated.

| Model | Full test | | UMLS-mapped | |
|---|---|---|---|---|
| | Macro F1 | Micro F1 | Macro F1 | Micro F1 |
| BERT-base ‡ | 1.34 | 15.81 | 1.45 | 18.80 |
| BART-base ‡ | 1.37 | 15.96 | 1.55 | 18.98 |
| ClinicalBERT ‡ | 1.53 | 16.32 | 1.76 | 19.46 |
| ClinicalBERT + CNN ‡ | 7.29 | 62.68 | 8.12 | 65.79 |
| GEN w/ BART | 31.15 | 78.90 | 31.22 | 81.27 |
| GEN w/ T5-base | 32.01 | 79.88 | 32.19 | 82.25 |
| Joint C + GEN | 32.89 | 79.93 | 32.70 | 82.97 |

Table 5: Pretrain-finetune performance. ‡ denotes classification methods. Models without symbols are generative. *Joint C + GEN* refers to both a classification and generative method (T5 + ClinicalBERT).

**Joint Classification & Generation.** To bring together the strengths of both discriminative and generative approaches within the pretrain-finetune framework, we implement a joint learning strategy. We use transformer-based models for both approaches, with shared transformer layers but distinct output layers for each task. This joint model structure is depicted in Figure 5.

**Results.** The generative techniques substantially outperformed the most effective string-matching method, demonstrating the power of such models. Leveraging T5 as the base model, the generative model achieved impressive micro F1 scores of $79.88\%$ and $82.25\%$ on the Full and UMLS-mapped test sets respectively. Interestingly, the multi-label classification didn't surpass the string matching approach, potentially due to the large class count (over 11K) and the bottleneck caused by the `[CLS]` token. However, incorporating a CNN on top of ClinicalBERT enhanced classification performance, reaching $62.68\%$ on the full set and $65.79\%$ on the UMLS-mapped set, but still lagging behind the generative model.

The joint learning model showed modest performance improvements, confirming the benefit of combined strategies.

The variance between micro and macro F1

| Model | Full test | | UMLS-mapped | |
| --- | --- | --- | --- | --- |
| | Macro F1 | Micro F1 | Macro F1 | Micro F1 |
| GPT-2 (Supervised) | 13.93 | 59.41 | 14.27 | 61.90 |
| ChatGPT (Zero-shot) | 24.69 | 39.75 | 22.23 | 38.10 |
| ChatGPT (One-shot) | 29.17 | 42.74 | 27.97 | 41.51 |

Table 6: Performance of prompting methods. While these methods fall short of the pretrain-finetune techniques, it is notable that the pretrain-finetune approach leverages the extensive SYMPTOMIFY dataset, underlining the considerable benefits of fine-tuning for improved results in settings with massive amounts of data such as in SYMPTOMIFY.

| Model | Rare symptoms test set | |
| --- | --- | --- |
| | Macro F1 | Micro F1 |
| String match † | 6.91 | 7.79 |
| GEN (T5-base) | 21.78 | 24.98 |
| GEN w/ definitions | 21.72 | **25.16** |
| GEN w/ synthetic reports | 23.31 | **25.73** |
| GPT-2 Prompting | 6.63 | 7.87 |
| ChatGPT Prompting (zero-shot) | 11.75 | 8.56 |
| ChatGPT Prompting (one-shot) | 16.06 | 10.49 |

Table 7: Macro F1 and Micro F1 of the symptom detection task on the rare symptoms test set.

scores in all configurations is due to the dataset's imbalance: 80% of symptoms are rare with 50 or fewer reports mentioning them, thus making it harder for the model to learn to accurately detect these infrequent symptoms.

### 5.4 Prompting Baselines

Given the widespread success of LM promoting, we also experimented with prompting baselines.

**Language Model Prompting.** We prompted ChatGPT using the following prompt: $x$ = *Extract a comma-separated list of symptoms from the following report:* "{*REPORT*}". In addition to zero-shot prompting, we explored the few-shot setting, specifically the one-shot setting with a single test set example added to the prompt.

Discriminative models assign mentions to specific class labels, while generative language models have a vocabulary that may differ from the formal symptom names in MedDRA. For instance, when generating the symptom "Pyrexia," ChatGPT often produces "Fever" instead. This necessitates answer engineering. We employ embedding-based methods using SBERT (Reimers and Gurevych, 2019). The prediction with the highest cosine similarity score between the predefined symptom set and the generated output is selected.

**Results.** Table 6 contains the results of the

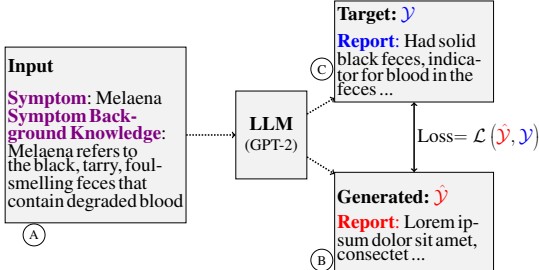

Figure 6: Synthetic Report Generation: the LM takes symptom names as input, and to encourage semantically faithful reports, also takes background knowledge as input (**A**), During training, target (**B**) and generated reports (**C**) are used to compute the loss.

prompting methods, and shows that they are unable to surpass the performance of the pretrain-finetune methods. This is expected since the pretrain-finetune models are trained on a large training data set of over 600K reports, whereas the zero-shot and one-shot prompting methods have access to zero or only one report, respectively. It is worth noting that the large-scale nature of SYMPTOMIFY highlights the significant benefits of fine-tuning for improved results in settings with massive amounts of data such as in SYMPTOMIFY.

### 5.5 Semantically Faithful Synthetic Data for Rare Symptoms

In our experiments, all methods struggled to identify rare symptoms. To address the rare symptoms problem, we used data augmentation techniques to generate more data points for these symptoms. We tried using symptom definitions as synthetic reports but found a stark discrepancy between the language in real reports and definitions. We instead used a simple synthetic data generation method to create more realistic data points for rare symptoms, illustrated in Figure 6. We trained a GPT-2 model on prompts comprising symptom name and definition. The definition is there to ensure that the report is semantically faithful to the meaning of the symptom, also reducing ambiguity in comparison to conditioning on the symptom name alone. Using Top-k multinomial sampling, we generated 100 synthetic reports per symptom, with optimal performance using 30 reports each. We then integrated this augmented data into our training set to improve the model's handling of rare symptoms.

**Results.** Table 7 illustrates the low performance achieved on the rare symptoms dataset, as indicated by both the macro and micro F1 scores across all

| (C1) Symptom | (C2) Report | (C3) Zero-shot Falcon Explanation | (C4) Falcon with SYMPTOMIFY Explanation |
|---|---|---|---|
| (R1) **Arthralgia** | Patient having shoulder pain. Made appointment with doctor. MRI completed. Possible SIRVA. | Arthralgia is mentioned in the patient report as it is a common symptom associated with SIRVA (Shoulder Impingement Syndrome with Radial Neuropathy) | The symptom "Arthralgia" is mentioned in the patient report because the patient reported shoulder pain. This is evident from the statement **"Patient having shoulder pain"**. |
| (R2) **Pyrexia** | all over body rash and fever | Pyrexia is mentioned in the patient report as it is a common symptom of fever | The symptom "Pyrexia" is mentioned in the patient report because the patient is experiencing a **fever**. |
| (R3) **Muscle spasms** | Patient stated he was getting muscle spasms in the neck and back after the flu shot. | Muscle spasms are a common side effect of the flu shot. The patient may have received the flu vaccine and experienced muscle spasms as a result. | The symptom "Muscle spasms" is mentioned in the patient report because the patient is experiencing muscle spasms in the neck and back after receiving a flu shot. |
| (R4) **Mood altered** | Fever, cold chills, restless sleep, redness at site of injection, right leg soreness-unable to stand on that leg, loss of appetite, moodiness/crying. After 12 hours of injection | The patient reported mood altered as a result of the injection. This is mentioned in the patient report as it is a potential side effect of the medication. | The symptom "Mood altered" is mentioned in the patient report because the patient reported experiencing a change in their mood. This is evident from the statement **"I was moody/crying."** |

Table 8: Falcon LM struggles to generate quality explanations (column **C3**) compared to ChatGPT. Yet, supplementing it with SYMPTOMIFY explanations improves its performance (column **C4**). Blue indicates accurate report fragments; red signifies paraphrased, non-exact matches, or hallucinations.

methods. Furthermore, we observe a modest improvement when incorporating synthetic reports into the training data. While the improvement is small, it demonstrates the (perhaps small) potential benefits of leveraging synthetic data to enhance the model's ability to handle rare symptoms.

## 5.6 Leveraging SYMPTOMIFY's Explanations in Falcon

We carried out an experiment to study the effectiveness of the explanations in SYMPTOMIFY using a prompting approach. For this, we used Falcon-7B-Instruct[8], a new, instruction-tuned iteration of the open-source Falcon renowned for its superior performance on key NLP benchmarks. The experiment had two configurations:
1) **Zero-shot Falcon Explanations:** We executed standard zero-shot in-context learning with Falcon.
2) **Falcon with SYMPTOMIFY Explanations:** We introduced few-shot learning in this setup, using selected SYMPTOMIFY explanations as our few-shots.

Table 8 provides a comparative analysis of Falcon's explanation generation under the two scenarios. Initially, Falcon struggled to produce beneficial explanations in the zero-shot learning context. However, its performance significantly improved in terms of specificity and context quoting when it was supplemented with few-shot learning, using a handful of SYMPTOMIFY explanations as a guide (contrast columns *C3* and *C4*, specifically *(R1,C4)*). Nonetheless, false quoting of hallucinated segments, as illustrated in *(R4,C4)* is a problem. To tackle this, one possible solution is to

[8] https://huggingface.co/tiiuae/falcon-7b-instruct

expose Falcon to a larger volume of SYMPTOMIFY explanations, thus improving its ability to generate high-quality explanations. However, due to the context size limitations in the prompting, the ideal solution to leverage these explanations effectively would be to implement the pretrain-finetune approach to use the massive SYMPTOMIFY dataset more productively as training data.

## 6 Conclusion

We introduced SYMPTOMIFY, a large-scale dataset of annotated reports, reflecting reactions to medications and vaccines. It includes MedDRA symptoms, annotation explanations, and background knowledge about symptoms, designed to facilitate development of systems that can aid human annotators operating in the critically important public health domain. Despite current performance on rare symptoms being low across all baselines, the continual evolution in language models and strides in few-shot learning offer promise for improvement.

There are several avenues for improving SYMPTOMIFY and expanding its utility. Integrating the explanations into a wider range of models may lead to the better generation of explained predictions with open source models such as Falcon. Expanding background knowledge to include relational knowledge, such as inter-symptom relationships, could yield performance gains. Further, focusing on laypeople's perspectives and acquiring more layman medical terms could facilitate a broader, more inclusive understanding of symptoms. Looking further ahead, as patient reports can sometimes contain not only textual descriptions but also ref-

erences to medical images, pictures submitted by patients, and lab test results, the incorporation of these non-textual elements using multi-modal language models, could be useful.

## Limitations

Our work, while it made useful contributions, has a number of significant limitations. First, the VAERS database, our primary data source, may not be representative of the larger population due to potential sampling biases. It also limits language diversity as it contains only English-language reports.

The symptom distribution in the reports presents a distinct challenge, with a strong skew towards certain symptoms that hinders model performance, including that of our most successful methods, like pretrain-finetune. While our attempts to rectify this through data augmentation techniques have yielded some improvement, the performance still falls short. This indicates the need for innovative methods that go beyond synthetic data expansion for addressing this challenge effectively.

The explanations present in SYMPTOMIFY are a rich resource that, as of now, we've only begun to explore. The explanations coupled with the public availability of the dataset offer a valuable base for other researchers to integrate into their work. Nevertheless, more exhaustive studies are needed to fully harness the potential of these explanations.

The potential biases of language models like ChatGPT, known to mirror the biases in the data they're trained on, is another limitation. Although we have not observed any explicit biases in our current study, no thorough testing for bias was conducted, and it remains a concern.

Finally, the use of non-medical experts from Amazon Mechanical Turk, as well as Computer Science graduate students for the annotation process introduces the possibility of inaccuracies into our assessment. Although we attempted to mitigate this by implementing a rigorous review process and utilizing multiple annotators, this factor may still have influenced the accuracy of our human evaluations. Future work may benefit from the involvement of medical experts in the annotation process, to increase the validity of our findings, and by extension the dataset.

## Ethics Statement

Models for symptom recognition have the potential to provide valuable insights for monitoring and as-

sessing the safety of medicines and vaccines. They can aid in the timely identification of potential risks and contribute to individual and public health. However, the availability of data to train such models, which is the main contribution of this work, raises concerns about individual privacy. Releasing the data, which includes free text reports and metadata from VAERS, may risk the exposure of personal information or identities. We rely on the privacy monitoring efforts of the CDC and FDA, the co-managers of VAERS. Furthermore, Following prior work on automated health systems, our goal is to be clear and transparent about the content of SYMPTOMIFY and the capabilities of the methods developed in this work.

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

# A Appendix

## A.1 Annotation Instructions for Mechanical Turk

We conducted three assessments via the Amazon Mechanical Turk (AMT) system and our annotation process. In this section, we describe detailed instructions and web interface screenshots used in the assessments.

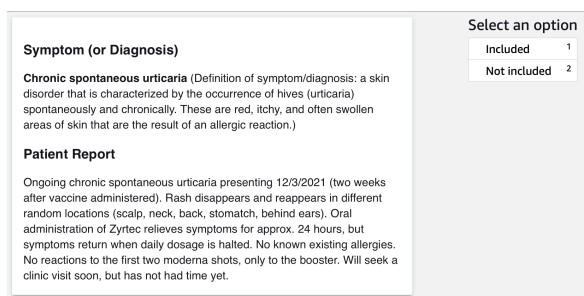

**Symptom (or Diagnosis)**

**Chronic spontaneous urticaria** (Definition of symptom/diagnosis: a skin disorder that is characterized by the occurrence of hives (urticaria) spontaneously and chronically. These are red, itchy, and often swollen areas of skin that are the result of an allergic reaction.)

**Patient Report**

Ongoing chronic spontaneous urticaria presenting 12/3/2021 (two weeks after vaccine administered). Rash disappears and reappears in different random locations (scalp, neck, back, stomatch, behind ears). Oral administration of Zyrtec relieves symptoms for approx. 24 hours, but symptoms return when daily dosage is halted. No known existing allergies. No reactions to the first two moderna shots, only to the booster. Will seek a clinic visit soon, but has not had time yet.

Select an option — Included 1 — Not included 2

Figure A.1: The AMT web interface used to evaluate the quality of MedDRA annotation in VAERS.

**1) VAERS MedDRA Annotation Quality Assessment**  As described in section 4.1, we evaluate the annotation quality of MedDRA annotations in VAERS. The instruction given to workers is "*Given a patient's report describes their medical conditions. The goal of this task is to check whether the given symptom is included in the report or not.*" For the detailed instruction, we provided the examples as below:

---

**Symptom/Diagnosis** : Anosmia
**Patient report** : 90% loss of smell/taste next day. Also feel irritable and nervous but manageable.
→ *The answer should be* **Included** *since the report mentions "90% loss of smell" in the report.*

---

**Symptom/Diagnosis** : Headache
**Patient report** : 90% loss of smell/taste next day. Also feel irritable and nervous but manageable.
→ *The answer should be* **Not included**

---

We also added the definition of the symptom in case that the workers are not familiar with the symptom (see Figure A.1).

**2) Explanation Quality Assessment**  The instruction given to workers is "*Given a patient report and the symptom, the explanation is describing why the*

---

[Pateint Report]: Patient was observed being unsteady ambulating with her cane. She was helped to a chair and assessed, alert times 4, slightly elevated blood pressure and respirations, and pulse of 60. Brief history was taken and she stated she was diabetic. Patient stated that she had taken her morning diabetic medication but had not eaten anything yet today. Patient was given a bottle of water, small bag of cookies, and a package of graham crackers. She was monitored continuously for 20 additional minutes at which time the client stated she felt fine and wanted to leave. She was walked to her vehicle and secured in her seat.

Explanation: The patient report mentions "Respiration abnormal" because the patient's respiration rate was slightly elevated as mentioned in the report, "alert times 4, slightly elevated blood pressure and respirations, and pulse of 60." The patient's medical history of being diabetic and taking medication without eating anything yet could have contributed to the abnormal respiration rate. However, the patient's condition improved after being given water and some snacks and was monitored constantly for 20 additional minutes before leaving.

Select an option — A: The explanation is both honest and helpful 1 — B: The explanation is honest but lacks details 2 — C: The explanation is neither honest nor helpful (i.e., incorrect explanation). 3

Figure A.2: The web interface used to evaluate the quality of explanations in SYMPTOMIFY.

*symptom is mentioned in the report. Choose* ***A*** *if the explanation is correct, and helpful in describing why the symptom is mentioned. Choose* ***B*** *if the explanation is correct, but lacks details, thus not helpful to explain why the symptom is mentioned. Otherwise choose* ***C***." Below are the examples we have provided for detailed instructions.

---

**Report** : Flu-like symptoms, extreme chills ( shivering ), sweats for 2 days
**Symptom** : Chill

---

**Explanation** : The patient report mentions "extreme chills (shivering)" as a symptom, referring to the presence of extreme chills or shivering experienced by the patient for 2 days.
→ *The answer should be* ***A***, *since the explanation is correctly describing as "The report mentions extreme chills (shivering) as a symptom" with a reason based on the report.*

---

**Explanation** : This report mentions presence of Chill.
→*The answer should be* ***B*** *even the explanation is correct, since the explanation is NOT helpful and too short to conjecture why the symptom is mentioned.*

---

**Explanation** : This report mentions presence of Chill. It states the patient experienced dizziness and shivering for 2 days.
→ *The answer should be* ***C*** *since the patient did NOT experience 'dizziness', i.e., the explanation is not correct.*

---

**3) GPT-3 Definition Quality Assessment**  The instruction given to workers is "*Given a term and definition, answer Yes or No whether the definition is correct for the term or not.*" We have provided the examples below for detailed instructions.

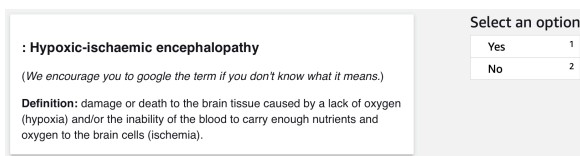

Term : Hypoxic-ischaemic encephalopathy

Select an option
Yes       1
No        2

(*We encourage you to google the term if you don't know what it means.*)

**Definition:** damage or death to the brain tissue caused by a lack of oxygen (hypoxia) and/or the inability of the blood to carry enough nutrients and oxygen to the brain cells (ischemia).

Figure A.3: The AMT web interface used to evaluate the quality of GPT-3 Definitions.

---

**Term** : Anosmia
**Definition** : Anosmia is a condition that results in a loss of the sense of smell.
→ *The answer should be **Yes***

---

**Term** : Headache
**Definition** : A reaction that occurs immediately following an injection.
→ *The answer should be **No***

---

We also added guidance to encourage annotators to Google search if they are unaware of the symptoms.

## A.2 Metadata Analysis

VAERS contains over 20 pieces of metadata, for example patient's basic information (e.g., age and sex), medical background (e.g., allergies, and medications), and vaccination information (e.g., vaccine type, and vaccine provider). We analyze this metadata. First, we observed that some symptoms occur more frequently in certain age groups, as shown in Figure A.4. The age group 0-9 is more likely to die after taking a vaccine than other groups. On the one hand, teenagers (age group 10-19) are more likely to get Syncope, Loss of Consciousness, and Pallor. Second, symptoms such as Paraesthesia Oral, and Migraine, are more likely to be observed in females than males. Syncope, Death, and Pallor are more common in males as shown in Figure A.5. Third, symptoms vary depending on the vaccine type. Headache and Pyrexia are the most common side-effects of COVID 19 vaccines, however, Diarrhoea and Haematochezia mostly occur after RV5 vaccination (see Table A.1).

## A.3 Experimental Setup Details

Table A.2 describes hyperparameters and search spaces we considered in experiments. For training transformer-based models, we used AdamW optimizer (Loshchilov and Hutter, 2017) for training all transformer-based models, and hyperparameters are set on the best validation performance. We ran experiments 3 times with different seeds, the reported scores are average of them. The results with

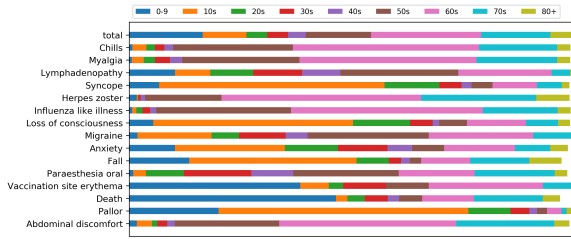

Figure A.4: Symptoms by patient age. Age group 0-9 is more likely to die after taking a vaccine.

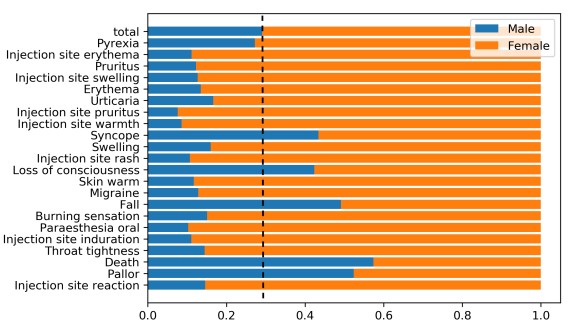

Figure A.5: Symptoms by patient sex. The dashed line denotes an average ratio of male to female, and shows that females report more. Symptoms such as Paraesthesia oral, and Migraine, are more likely to be observed in females than males.

standard deviations are presented in Table A.4.

For generative models, we adopted GENRE's (Cao et al., 2021) experimental settings with 256 of maximum input length, 128 of maximum output length, 64 of batch size, and 4 of beam search size. We used the pre-trained BART-base (Lewis et al., 2020) with 3e-5 of learning rate and T5-base (Raffel et al., 2019) with 1e-4 of learning rate following the original papers. We fine-tuned each model 5 epochs on our training set.

**Evaluation of Generative Models** While discriminative models can identify exact classes, generative models cannot predict discrete classes. In our work, we form the target text as a sequence of symptoms separated by ";". We split the generated text by "," and match each chunk with symptom names by exact string match. For those symptoms that are not exactly matched within our symptom set, we post-processed answer engineering as described in Section 5.4.

**Evaluation of MetaMap** We use Metamap for one of zero-shot baseline models. Given the input text, MetaMap outputs UMLS entities with confidence scores. We experimented with thresholds

| | COVID-19 | FLU4 | HPV9 | TDAP | RV5 |
|---|---|---|---|---|---|
| 1 | Headache | Pain | Syncope | Injection site pain | Diarrhoea |
| 2 | Pyrexia | Injection site pain | Dizziness | Pain | Haematochezia |
| 3 | Fatigue | Pain in extremity | Headache | Pain in extremity | Vomiting |
| 4 | Chills | Injection site erythema | Injection site pain | Injection site swelling | Intussusception |
| 5 | Pain | Pyrexia | Nausea | Injection site erythema | Pyrexia |

Table A.1: Top 5 most frequent symptoms by vaccine type.

| Metamap | |
|---|---|
| threshold | 0.1 |
| | {0.05, 0.1, 0.15, 0.2, 0.25} |
| **Multi-label Classification** | |
| Learning rate | 5e-5 {2e-5, 3e-5, 5e-5} |
| max input length | 256 {128, 256} |
| batch size | 32 |
| threshold | 0.1 |
| | {0.05, 0.1, 0.15, 0.2, 0.25} |
| **Generative Models** | |
| Learning rate (BART-base) | 3e-5 {2e-5, 3e-5, 5e-5} |
| Learning rate (T5-base) | 1e-4 {3e-5, 5e-5, 1e-4, 2e-4} |
| max input length | 256 {128, 256} |
| max output length (decoder) | 128 {64, 128, 256} |
| beam search size | 4 |
| batch size | 64 {32, 64} |
| **Joint Learning (C + GEN)** | |
| Learning rate | 3e-5 {2e-5, 3e-5, 5e-5} |
| max input length | 256 {128, 256} |
| max output length (decoder) | 128 |
| batch size | 64 {32, 64} |
| $\lambda_j$ | 0.1 {0.05, 0.1, 0.2, 0.5} |
| **LM Prompting (GPT-2)** | |
| Learning rate | 5e-5 {2e-5, 3e-5, 5e-5} |
| max input length | 512 |
| batch size | 64 {32, 64} |

Table A.2: Best-performing hyperparameters and search space. Values in parentheses denote search space.

| | Symptom |
|---|---|
| 1 | Headache |
| 2 | Pyrexia |
| 3 | Fatigue |
| 4 | Chills |
| 5 | Pain |
| 6 | Nausea |
| 7 | Pain in extremity |
| 8 | Dizziness |
| 9 | Injection site pain |
| 10 | Myalgia |

Table A.3: Top 10 symptoms in SYMPTOMIFY

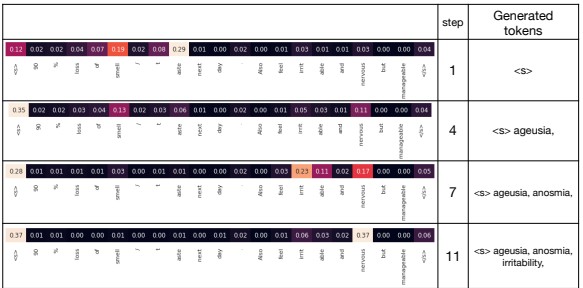

(a) Attention map according to the decoding step of the generative model

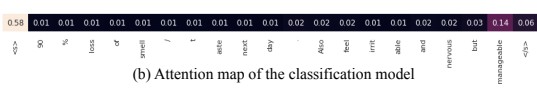

(b) Attention map of the classification model

Figure A.6: Generation vs Classification: Attention patterns in each approach.

{0.05, 0.1, 0.15, 0.2, 0.25, 0.3} and regarded entities as positives over the threshold. The threshold of 0.1 was determined on the best validation score.

**Computing Infrastructure** We ran all experiments on a single NVIDIA GTX 108 (12GB) with CUDA 10.1 version.

## A.4 More Analysis

**Generation vs Classification** We can observe the significant gap in performance between classification and generation models. Figure A.6 shows the difference of attention patterns in each approach. In the generative model, the decoder attends input tokens that are related to symptoms at each decoding step. On the other hand, the classification model uses one aggregated vector to predict all symptom classes. We can conclude that considering most examples have multiple symptoms, predicting at once is not sufficient to recall multiple entities.

**Negation Performance** Due to the prevalence of complex negation in our dataset, we analyzed how the model performs in the presence of negation. We sampled 103 examples of those patterns from the development set and tested them with different training sizes. Figure A.7 presents performance

| Model | Full test | | UMLS-mapped | |
| --- | --- | --- | --- | --- |
| | Macro F1 | Micro F1 | Macro F1 | Micro F1 |
| Exact String Matching | 8.87 | 30.38 | 10.41 | 32.98 |
| MetaMap | - | - | 3.74 | 10.65 |
| Symptom Embeddings | 0.72 | 3.32 | 0.87 | 5.87 |
| BERT-base ‡ | 1.34 (±0.13) | 15.81 (±0.14) | 1.45 (±0.09) | 18.80 (±0.08) |
| BART-base ‡ | 1.37 (±0.18) | 15.96 (±0.17) | 1.55 (±0.12) | 18.98 (±0.13) |
| ClinicalBERT ‡ | 1.53 (±0.12) | 16.32 (±0.12) | 1.76 (±0.11) | 19.46 (±0.11) |
| ClinicalBERT + CNN ‡ | 7.29 (±0.35) | 62.68 (±0.25) | 8.12 (±0.47) | 65.79 (±0.25) |
| GEN w/ BART † | 31.15 (±0.37) | 78.90 (±0.24) | 31.22 (±0.19) | 81.27 (±0.25) |
| GEN w/ T5-base † | 32.01 (±0.31) | 79.88 (±0.27) | 32.19 (±0.22) | 82.25 (±0.25) |
| Joint C + GEN | **32.89** (±0.21) | **79.93** (±0.19) | **32.70** (±0.43) | **82.97** (±0.31) |
| GPT-2 (Supervised) | 13.93 | 59.41 | 14.27 | 61.90 |
| ChatGPT (Zero-shot) | 24.69 | 39.75 | 22.23 | 38.10 |
| ChatGPT (One-shot) | 29.17 | 42.74 | 27.97 | 41.51 |

(Rows BERT-base through Joint C + GEN are grouped under "Pretrain-Finetune"; GPT-2, ChatGPT rows are grouped under "Prompt".)

Table A.4: Performance summary. ‡ denotes classification methods and † denotes generative methods. *Joint C + GEN* refers to both a classification and generative method (T5 + ClinicalBERT).

| Type | Example |
| --- | --- |
| no [symptom name] | I feel cold, but do not have a fever, am not having chills, nausea, etc. |
| | Fever of 101, 102 for 2 Days; Chills; Nausea with no Vomiting. |
| | no visible signs of rash or irritation. |
| Medical history (not current symptom) | Medical history included Ankylosing, spondylitis/psoriasis, cardiac ablation from Jan2017 |
| | Family medical history included her dad had rheumatic fever and high blood pressure. |
| | Past medical history included no adverse event. concurrent medical conditions included Ulcer and Hypothyroidism. |
| Other negation expressions | Patient didn't experience fever and rash, but feel muscle pain |
| | All but the pain and soreness at the injection site subsided by the next day |
| | Denies any further fevers or chills. Denies any nausea vomiting. Denies loss stasis smell. |

Table A.5: Examples of negation patterns in SYMPTOMIFY

on 103 negation samples as we varied the number of train examples. As the size of the training set increases, the model performs better on negation expressions and the performance saturates at $200k$. Through this result, we can conclude that our dataset contains sufficient negation data points and effectively learns to deal with a large number of them.

We observed that the simple negation expression "no [symptom name]" is indeed an easy form of negation, the model performs well on it, even with a small amount of train set around $50k$ examples. A more thorough analysis of which negation expressions the model still fails on after saturation, is left as future work. Table A.5 shows examples of negation expressions in SYMPTOMIFY.

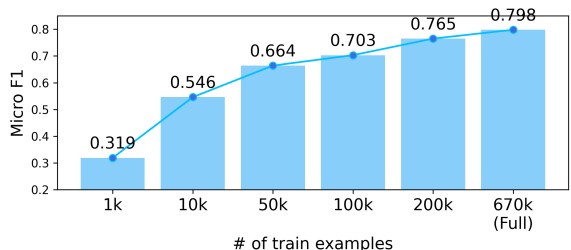

Figure A.7: Micro F1 of the generative model with T5-base on data points involving negation as the size of the training data is varied. As the size of the training set increases, the model performs better on negation expressions and the performance saturates around $200k$.