# OpenReview forum: "SYMPTOMIFY: Transforming Symptom Annotations with Language Model Knowledge Harvesting"
_EMNLP/2023/Conference — EMNLP 2023 Findings_

### Official Review · Reviewer_jY5d · 2023-08-04

**Soundness:** 3

**Ethical Concerns:**

Yes

**Excitement:**

3: Ambivalent: It has merits (e.g., it reports state-of-the-art results, the idea is nice), but there are key weaknesses (e.g., it describes incremental work), and it can significantly benefit from another round of revision. However, I won't object to accepting it if my co-reviewers champion it.

**Paper Topic And Main Contributions:**

The paper introduces SYMPTOMIFY, a dataset of annotated vaccine adverse reaction reports that aims to improve the efficiency of human annotators in healthcare decision-making. The dataset contains over 800k reports, reasoning-based explanations, and background knowledge obtained from language model knowledge harvesting. The paper evaluates various baselines across different learning paradigms and provides insights for future comparisons and benchmarking.
**Contribution:**
- The paper addresses a significant problem in healthcare decision-making and proposes a dataset that can aid human annotators.
- The dataset is comprehensive, featuring a large number of reports and including explanations and background knowledge.
- The paper provides insights and evaluation results for various baselines and learning paradigms, which can guide future research and comparisons.

**Questions For The Authors:**

1. The effectiveness and accuracy of harvesting background knowledge from LLM are uncertain. Are there problems such as hallucinations that lead to the wrong knowledge? I think these tasks require professionals to annotate.
2. The results of different baselines are placed in different tables, so it is not easy to compare with each other.
3. Some of the experimental results appear to be very poor, with some F1 scores less than 2.0 in Table 4 & 5.


**Reasons To Accept:**

1. The article introduces a new method using language model knowledge to improve the SYMPTOMIFY dataset. This method expands existing research on symptom recognition and has the potential to advance medical decision-making.
2. By incorporating LLM, the article improves the SYMPTOMIFY dataset by adding explanations and background information. This enhancement provides valuable context and detailed information for symptom recognition, improving the overall quality and usefulness of the dataset.

**Reasons To Reject:**

1. The evaluation of the proposed method is limited, and it would be beneficial to have more extensive evaluation and ablation studies.
2. The idea of validating the quality of annotations is good, but also needs to be done by professionals, not crowdsourcing. Therefore, the verification results cannot be trusted.

**Reproducibility:**

3: Could reproduce the results with some difficulty. The settings of parameters are underspecified or subjectively determined; the training/evaluation data are not widely available.

**Reviewer Confidence:**

3: Pretty sure, but there's a chance I missed something. Although I have a good feel for this area in general, I did not carefully check the paper's details, e.g., the math, experimental design, or novelty.

**Typos Grammar Style And Presentation Improvements:**

The size of Figure 2 and Figure 3 can be adjusted to be consistent.

---

> ### Author Rebuttal · Authors · 2023-08-27
>
> Thank you for your thorough, and constructive review of our work. We are committed to addressing the concerns and suggestions you've raised to improve the clarity and quality of our work.
>
> **Scope of evaluation and ablations.** We agree that comprehensive experiments are a stronger contribution.  We implemented and evaluated various baselines across different methods and learning paradigms,
> *i) String Matching*: Involves matching symptom names directly with the report text based on discrete symbols.
> *ii) Symptom Embeddings*: Represents symptom names and report text as dense vectors for matching in vector space.
> *iii) Pretrain-Finetune*: Uses pre-trained language models fine-tuned on our dataset.
> *iv) Prompting*: Leverages prompts to generate responses from language models in zero-shot and few-shot settings.
>
> We believe this variety of  baselines facilitates future comparisons and benchmarking.
>
> **Poor Experimental Results for Certain Baselines.** Our analysis showed that the main factor contributing to poor results is the long-tail nature in the dataset. Since most symptoms (> 80%) are rare.  Another factor influencing the low F1 scores  in classification baselines (in contrast to generative methods) is the bottleneck created by the classifier layer commonly employed in the classification setup of Language Model models (LLMs). Typically, the classification model utilizes a single aggregated vector (such as the [CLS] vector in BERT) along with an additional classifier layer positioned atop the LLM architecture. Following this practice, in our task setup, we rely on only one aggregated vector to predict outcomes across all symptom classes (a total of 11,472 classes). It becomes apparent that this approach is not sufficient to predict that many entities at once, considering that most examples within SYMPTOMIFY involve multiple symptoms. We present an analysis of this issue, outlining the notable disparity of attention patterns between classification and generation models, in Appendix A.4. We will clarify these statements in the main paper body.
>
> **One table for different baselines.** Our intention was to present the results through various approaches: the 'zero-shot approach' (Table 4), the 'Pretrain-finetune approach' (Table 5), and the 'Prompting approach' (Table 6). Additionally, we provide separate results for the Rare symptom test set (Table 7), which are results on a different test set unlike the others. To enhance readability, we will include a comprehensive table that allows for a consolidated view of all comparisons in a single place.
>
> **Types of annotators used.** The VAERS dataset has been initially annotated by trained professional MedDRA coders. Our supplementary validation relied on perspectives of three distinct groups who inspected the  VAERS annotations across varied tasks: i) crowdworkers,  ii) ChatGPT, and iii) graduate students that have experience with biomedical entity linking.   Across these annotation tasks, we observed agreement rates similar to related annotation tasks reported in the literature which showed agreements in the range: 0.55-0.62 (Wadhwa et al., 2022; Nye et al., 2018; Deleger et al., 2012). When we manually inspected the annotation from crowdworkers, the majority of errors come  from medical test results not explicitly mentioned in patient reports (such as C-reactive protein levels or metabolic function tests). Thus, although we lacked access to trained MedDRA coders, we adhered to (and went further than) the established standards in NLP annotation protocols. We will clarify this in the paper.
>
> **The quality of harvested knowledge from LLM.** The explanations we harvested from the LLMs are encouraged to cite segments of mentions from the patient reports, thus this grounding of mentions of symptoms discourages hallucination. We found that ChatGPT avoided hallucination  in explanations especially with prompts that included this grounding requirement, whereas the open source, smaller Falcon LLM  struggled with hallucinations even with prompts that required grounding; in the paper, we  provided examples of Falcon hallucinations of spans in Table 8 on page 8.
>
> **Sizes of Figures 2 and 3:** Thank you for this presentation improvement comment, we will change their sizes to be consistent. Figure 2 has less content, thus it is currently smaller, but we agree that the current presentation and contrast with the much larger Figure 3 is not ideal.

---

### Official Review · Reviewer_7Q9D · 2023-08-04

**Soundness:** 3

**Excitement:**

4: Strong: This paper deepens the understanding of some phenomenon or lowers the barriers to an existing research direction.

**Paper Topic And Main Contributions:**

In this paper, the authors extend the VAERS dataset with annotations of symptom information with the help of ChatGPT. In additional, the authors assess the quality and utility of the new dataset by performining the task of symptom recognition using a variety of approaches.

**Reasons To Accept:**

- The new dataset is large (800k entries) and would be an additional resource for the community.

- The experiments include interesting and very up-to-date attempts, e.g. addressing rare symptoms, addressing the zero-shot scenario, and incorporating the Falcon-7B-Instruct model.

**Reasons To Reject:**

- The VAERS dataset already contains MedDRA codes of symptoms. The new dataset provides textual spans and additional information of the symptoms. In my opinion, the added value is limited.

- The annotations of the VAERS dataset are trained professionals. It is a good idea to verify the quality of the annotations, but this needs to be done by professionals as well, not crowd sourcing workers. Therefore, I consider the results of this verification untrustworthy.

- Some of the experimental results are very poor, e.g. F1 scores as low as 1.34! Something seems wrong (model setting? bug in code?) and urgently needs further explanation.

**Reproducibility:**

4: Could mostly reproduce the results, but there may be some variation because of sample variance or minor variations in their interpretation of the protocol or method.

**Reviewer Confidence:**

3: Pretty sure, but there's a chance I missed something. Although I have a good feel for this area in general, I did not carefully check the paper's details, e.g., the math, experimental design, or novelty.

**Typos Grammar Style And Presentation Improvements:**

The title is misleading. "Knowledge harvesting" refers to the established task of mining large amounts of text to create a knowledge base. The way ChatGPT is used in this paper would be better characterized as "leveraging knowledge in a large language model."

---

> ### Author Rebuttal · Authors · 2023-08-27
>
> We appreciate your constructive comments and questions. We are committed to addressing the concerns and suggestions you've raised to improve the clarity and quality of our work.
>
> **Scope of Contributions.** Our annotations of explanations and background knowledge complement MedDRA codes by providing additional context and supporting human coders in decision-making during the annotation process. Furthermore, our annotations facilitate the development of symptom recognition models capable of both prediction and explanation, a capability unavailable in the original VAERS dataset. Given the critical role of explainability in healthcare, we believe our contributions add value. We will state this with more clarity in the paper.
>
> **Annotations Verification Process.** The VAERS dataset has been initially annotated by trained professional MedDRA coders. Our supplementary validation relied on perspectives of three distinct groups who inspected the  VAERS annotations across varied tasks: i) crowdworkers, ii) ChatGPT, and iii) graduate students that have experience in biomedical entity linking. Across these annotation tasks, we observed agreement rates similar to related annotation tasks reported in the literature which showed agreements in the range: 0.55-0.62 (Wadhwa et al., 2022; Nye et al., 2018; Deleger et al., 2012). When we manually inspected the annotation from crowdworkers, the majority of errors come  from medical test results not explicitly mentioned in patient reports (such as C-reactive protein levels or metabolic function tests). Thus, although we lacked access to trained MedDRA coders, we adhered to (and went further than) the established standards in NLP annotation protocols.  We will clarify this in the paper.
>
> **Poor Experimental Results for Certain Baselines.** Our analysis showed that the main factor contributing to poor results is the long-tail nature in the dataset. Since most symptoms (> 80%) are rare.  Another factor influencing the low F1 scores  in classification baselines (in contrast to generative methods) is the bottleneck created by the classifier layer commonly employed in the classification setup of Language Model models (LLMs). Typically, the classification model utilizes a single aggregated vector (such as the [CLS] vector in BERT) along with an additional classifier layer positioned atop the LLM architecture. Following this practice, in our task setup, we rely on only one aggregated vector to predict outcomes across all symptom classes (a total of 11,472 classes). It becomes apparent that this approach is not sufficient to predict that many entities at once, considering that most examples within SYMPTOMIFY involve multiple symptoms. We present an analysis of this issue, outlining the notable disparity of attention patterns between classification and generation models, in Appendix A.4. We will clarify these statements in the main paper body.
>
> **Title Refinement.** We appreciate your suggestion for refining the title to accurately represent our paper's content. We will update the title to reflect this characterization of "Leveraging Knowledge in a Large Language Model,".

---

### Official Review · Reviewer_Y8AL · 2023-08-12

**Typos Grammar Style And Presentation Improvements:** Line 427
**Soundness:** 4

**Excitement:**

4: Strong: This paper deepens the understanding of some phenomenon or lowers the barriers to an existing research direction.

**Paper Topic And Main Contributions:**

This paper is about aiding symptom annotation by taking advantage of the LMs. The main contribution is the symptomify dataset having medical reports, annotation explanations using LM knowledge harvesting and background information regarding symptoms. The authors also explore performance of various baseline models on this data.


**Questions For The Authors:**

A. Line 158: It is not clear where this specific number of symptoms come from? Does the dataset have 1000 symptom texts or is this randomly chosen?
B. What is the agreement between  Chat-GPT as the 4th annotator and the other 3 human annotators?
C. Line 172: What are these 1000 tasks?
D. Line 214: Are these 1000 samples statistically significant? Same with Line 275.


**Reasons To Accept:**

1.Motivation and background is well written. The work is also nicely situated in the relevant areas of research.
2.The LM knowledge harvesting is interesting in this context
3.Overall, the authors put forward an approach which utilizes the power of LMs and make AI accessible. The thought of using AI as a tool, rather than a human replacement is commendable.


**Reasons To Reject:**

The annotation process of the Symptomify dataset is not clear. Since the main contribution of the paper is the dataset, the correctness of the annotation can be an issue. Also, there are some unanswered questions (Please see Questions for Authors).

**Reproducibility:**

4: Could mostly reproduce the results, but there may be some variation because of sample variance or minor variations in their interpretation of the protocol or method.

**Reviewer Confidence:**

3: Pretty sure, but there's a chance I missed something. Although I have a good feel for this area in general, I did not carefully check the paper's details, e.g., the math, experimental design, or novelty.

---

> ### Author Rebuttal · Authors · 2023-08-27
>
> Thank you for your valuable comments and thoughtful review. We are fully committed to incorporating these suggestions to improve the clarity and quality of our paper.
>
> **In relation to points A and C (choice of evaluated patient reports):**
> SYMPTOMIFY has 839,215 patient reports  (symptom texts). To carry out our analysis, we selected 1000 symptom **names** at random. For each of these symptom names, we also randomly selected a corresponding patient report (symptom text) that mentions that specific symptom. Our objective was to incorporate a broad spectrum of symptoms within our evaluations, thereby avoiding any inclination towards analyzing only a skewed subset of patient reports that might disproportionately focus on common symptoms like headache or pyrexia. We have included  instructions and screenshots in the Appendix to offer further details of our methodology, and we will clarify these statements in the main paper body.
>
> **In relation to point B (agreement between ChatGPT and human annotators).** The agreement between ChatGPT and the 3 human annotators is 0.54 Fleiss' Kappa representing moderate agreement rate that is similar to related annotation tasks reported in the literature where  agreements fell within the range: 0.55-0.62 (Wadhwa et al., 2022; Nye et al., 2018; Deleger et al., 2012). Note that these prior settings differ from our setting because they do not include ChatGPT as a fourth annotator as we did here. We will clarify these details in the paper.
>
> **In relation to point D (Evaluation sample size).**
> In terms of statistical significance, we evaluated a set of 1000 randomly selected symptoms from the total pool of 11,472 symptoms; this represents over 8% of the entire symptom set  within the SYMPTOMIFY dataset. This subset serves as a representative sample for our analysis, allowing us to draw meaningful conclusions.
>
> **Others (typos & presentation):** Thank you for this presentation improvement comment, regarding the typo on line  427.

---

### Meta-Review · Area_Chair_xTAt · 2023-09-17

**Recommendation:** 4

**Metareview:**

**Summary:**
In this paper, the authors introduce SYMPOMIFY, an extension of the VAERS dataset that includes annotations of symptom information. These annotations were generated using Large Language Model (LLM) knowledge harvesting, particularly with the assistance of ChatGPT. The dataset comprises over 800,000 reports, along with reasoning-based explanations and background knowledge. Additionally, the authors evaluate the quality and utility of the new dataset by conducting symptom recognition tasks across various learning paradigms, providing insights for future comparisons and benchmarking.

**Strengths:**
The reviewers unanimously agree on the following strengths:
1. The article introduces the SYMPTOMIFY dataset which incorporates explanations and background information, offering valuable context and detailed data for symptom recognition, thereby improving the overall quality and utility of the dataset.
2. The newly introduced dataset in this work is notably substantial, with over 800,000 entries, providing an additional and valuable resource for the research community.
3. The experiments conducted in the paper are both interesting and up-to-date, including efforts to address rare symptoms, address zero-shot scenarios, and incorporate the Falcon-7B-Instruct model.
4. The method outlined by the authors expands the existing field of the research on symptom recognition and holds the potential to advance medical decision-making.
5. The paper offers valuable insights and evaluation results for various baselines and learning paradigms, serving as a guide for future research and comparisons.
6. The authors' approach harnesses the capabilities of Large Language Models (LLMs) to make AI an accessible tool, rather than a human replacement.

**Weaknesses:**
All the reviewers share the same concerns regarding this work.
First and foremost, the annotations in the VAERS dataset are carried out by trained professionals. While it's a commendable idea to assess the quality of these annotations, it should ideally be conducted by experts rather than crowdsourced workers. Consequently, the results of the verification in this work may lack reliability. Given that the primary contribution of the paper is the dataset itself, the accuracy of the annotations becomes a critical concern.
Additionally, the evaluation of the proposed method is somewhat limited, and it would be advantageous to have a more extensive evaluation and conduct ablation studies.

**Author-Reviewer discussion and acknowledgment:**
The authors have provided clarifications in response to the concerns raised by the reviewers and have outlined the planned improvements to be made during the rebuttal response and discussion phase. All reviewers have responded and acknowledged the authors' arguments.

**Conclusion:**
The paper is well-motivated, and the background is well-written. The work is also appropriately contextualized within the relevant areas of research. However, reviewers suggest that the authors correct the identified typos. Furthermore, reviewers recommend that the authors improve the paper by addressing the questions and points raised during the discussion phase.

---

### Decision · Program_Chairs · 2023-10-07

**Decision:**

Accept-Findings

**Comment:**

**Summary:**
In this paper, the authors introduce SYMPOMIFY, an extension of the VAERS dataset that includes annotations of symptom information. These annotations were generated using Large Language Model (LLM) knowledge harvesting, particularly with the assistance of ChatGPT. The dataset comprises over 800,000 reports, along with reasoning-based explanations and background knowledge. Additionally, the authors evaluate the quality and utility of the new dataset by conducting symptom recognition tasks across various learning paradigms, providing insights for future comparisons and benchmarking.

**Strengths:**
The reviewers unanimously agree on the following strengths:
1. The article introduces the SYMPTOMIFY dataset which incorporates explanations and background information, offering valuable context and detailed data for symptom recognition, thereby improving the overall quality and utility of the dataset.
2. The newly introduced dataset in this work is notably substantial, with over 800,000 entries, providing an additional and valuable resource for the research community.
3. The experiments conducted in the paper are both interesting and up-to-date, including efforts to address rare symptoms, address zero-shot scenarios, and incorporate the Falcon-7B-Instruct model.
4. The method outlined by the authors expands the existing field of the research on symptom recognition and holds the potential to advance medical decision-making.
5. The paper offers valuable insights and evaluation results for various baselines and learning paradigms, serving as a guide for future research and comparisons.
6. The authors' approach harnesses the capabilities of Large Language Models (LLMs) to make AI an accessible tool, rather than a human replacement.

**Weaknesses:**
All the reviewers share the same concerns regarding this work.
First and foremost, the annotations in the VAERS dataset are carried out by trained professionals. While it's a commendable idea to assess the quality of these annotations, it should ideally be conducted by experts rather than crowdsourced workers. Consequently, the results of the verification in this work may lack reliability. Given that the primary contribution of the paper is the dataset itself, the accuracy of the annotations becomes a critical concern.
Additionally, the evaluation of the proposed method is somewhat limited, and it would be advantageous to have a more extensive evaluation and conduct ablation studies.

**Author-Reviewer discussion and acknowledgment:**
The authors have provided clarifications in response to the concerns raised by the reviewers and have outlined the planned improvements to be made during the rebuttal response and discussion phase. All reviewers have responded and acknowledged the authors' arguments.

**Conclusion:**
The paper is well-motivated, and the background is well-written. The work is also appropriately contextualized within the relevant areas of research. However, reviewers suggest that the authors correct the identified typos. Furthermore, reviewers recommend that the authors improve the paper by addressing the questions and points raised during the discussion phase.